# Improved birth rates via rehydration of mouse freeze-dried spermatozoa using high-temperature ultrapure water

**Kango Yamaji[1], Sayaka Wakayama[2], Natsuki Ushigome[1], Daiyu Ito[1], Teruhiko Wakayama[1,2]\***

**1** Faculty of Life and Environmental Science, University of Yamanashi, Kofu, Japan, **2** Advanced Biotechnology Center, University of Yamanashi, Kofu, Japan

\* twakayama@yamanashi.ac.jp

## Abstract

Freeze-drying (FD) is a promising method for achieving the long-term, low-cost, and safe preservation of mammalian sperm at room temperature (RT, 23–25°C). However, the birth rate of embryos fertilized with FD sperm is reduced to less than half compared to those fertilized with fresh sperm. Moreover, the underlying causes and potential solutions remain unclear. In this study, we investigated a rehydration process using FD sperm to determine its effects on sperm DNA damage. We also attempted to optimize this rehydration method to improve birth rates. We initially examined the effects of slowing water infiltration into mouse FD sperm using a high osmolarity or viscosity solution, but we found that this increased DNA damage and decreased birth rates. Next, to accelerate infiltration speed, we performed rehydration of FD sperm using ultrapure water heated up to as hot as 90°C. However, we found that the DNA damage of the FD sperm decreased as the temperature increased. The level of DNA damage in the male pronucleus at the zygote stage and of abnormal chromosome segregation (ACS) at the two-cell stage were also decreased at 37°C or 50°C. Finally, the birth rates of embryos derived from FD sperm also significantly improved when rehydration was performed using 50°C ultrapure water (37%) compared with the RT control (22%). Taken together, the results of this study demonstrate that the DNA of FD sperm can be damaged during the rehydration process and that rapid rehydration significantly improves the birth rate.

## Introduction

Mammalian gametes are preserved for a variety of purposes, including infertility treatments, the conservation of endangered species, the cost-effective storage of genetically modified organisms, and the efficient transportation of mouse strains [1–3]. The standard method for preserving mammalian spermatozoa is cryopreservation using

**Data availability statement:** All relevant data are within the manuscript and its Supporting information files.

**Funding:** This work was partially funded by JST SPRING, Grant Number JPMJSP2133 to N. U.; the Research Fellowships of Japan Society for the Promotion of Science for Young Scientists to D.I. (23K19330), S. W. (23K08843) to T.W. (23K18124 and 24K01779); the Naito Foundation and Takahashi Industrial and Economic Research Foundation (189) to S.W.; Asada Science Foundation and the Canon Foundation (M20-0008) to T.W.

**Competing interests:** The authors have declared that no competing interests exist.

liquid nitrogen [4]. This technique maintains sperm motility post-thaw, which makes it suitable for *in vitro* fertilization (IVF) [5]. However, handling liquid nitrogen is challenging and poses risks of frostbite, asphyxiation, and rapid gas expansion. Moreover, it is also necessary to frequently replenish liquid nitrogen to keep temperatures low, and specialized containers are required for storage and transport, leading to high maintenance costs [1,2]. Furthermore, during disruptions of the liquid nitrogen supply (e.g., during natural disasters) this poses a risk of losing all stored samples [6].

Freeze-drying technologies offer a promising solution to simultaneously address the multiple challenges associated with cryopreservation. Since 1998, when the first successful freeze-drying of mouse sperm was demonstrated [7], viable offspring have been produced from embryos fertilized with freeze-dried (FD) sperm from rats [8], hamsters [9], rabbits [10], and horses [11]. Initially, the room temperature (RT) storage period of FD sperm was limited to only one month [7]. However, by sealing FD sperm in ampoules under a high vacuum, sperm storage at RT for longer periods became possible, with some studies reporting successful storage for periods of up to six years [12,13]. This technology eliminates the need for liquid nitrogen and permits storage in compact spaces such as desk drawers [14]. Overall, this technique enables space-efficient and cost-effective storage while reducing the risk of sample loss during disasters. In addition, the development of safer and more user-friendly storage containers, such as plastic sheets [15], microtubes [16], and mini stainless steel tubes [17], is currently underway. Furthermore, sperm subjected to FD treatment showed improved resistance to extreme temperatures ranging from −196°C to 150°C [18], and to extreme radiation. FD sperm stored on the International Space Station showed no reduction in fertilization capacity or developmental potential after 6 years of storage, despite exposure to high levels of radiation [19,20]. Taken together, these findings highlight the advantages of FD preservation over cryopreservation for the long-term storage of sperm. However, in the mouse, the birth rates of embryos derived from FD sperm are less than one-third of those of embryos derived from fresh or cryopreserved sperm [13,21–23]. Optimal cryopreservation methods vary by species, resulting in interspecies differences in FD sperm tolerance and DNA stability [24,25]. However, birth rates remain extremely low across all animal species in which offspring have been obtained from FD sperm. Thus, for FD methods to become more widely used for preserving genetic resources, it is essential to improve the rate at which they reliably produce offspring from preserved sperm. In animals such as cattle, which produce only one offspring per pregnancy, low birth rates necessitate multiple attempts, which can result in high overall costs despite improvements in sperm storage efficiency and safety.

Embryos fertilized with FD sperm frequently exhibit both DNA damage in the paternal pronucleus and chromosomal segregation abnormalities (ACS) [26]. These findings suggest that sperm DNA damage occurs at some stages of FD sperm preparation or use. Preparing and using FD sperm is a process that consists of four steps: freezing, drying, storage at RT, and rehydration of FD sperm. During the freezing, cryoprotectants such as DMSO or glycerol cannot be used because they cannot be

dried, therefore all the sperm die after freezing. However, intracytoplasmic sperm injection (ICSI) allows injection of all sperm—including dead sperm—into oocytes [27], thereby achieving normal fertilization and full-term development. This indicates that decreases in the birth rate of FD sperm are not due to the freezing step [22,28,29]. To investigate the effect of the drying process, we conducted an experiment in which sperm were directly vacuum-dried without freezing. This experiment demonstrated that viable offspring could be obtained, although they showed lower birth rates than FD sperm [30]. Taken together, these findings suggest that damage to FD sperm primarily occurs during drying, and that freezing may protect sperm from drying-associated DNA damage.

During RT storage, damage to FD sperm is primarily caused by air contamination. Therefore, storing ampoules under a high vacuum reduces damage to FD sperm and enables long-term RT preservation [12]. Interestingly, although exogenous trehalose supplementation of the storage medium did not reduce sperm damage during freezing and drying, it significantly improved birth rates after long-term storage of FD sperm at RT [14]. This finding indicates that reduced birth rates of embryos derived from FD sperm following preservation at RT results not only from damage to FD sperm during freezing and drying but also from additional damage sustained during RT storage.

In this study, we focused on the rehydration of FD sperm, a step that has not yet been thoroughly investigated. To do so, we prepared three types of solutions with varying osmotic pressures, viscosities, and temperatures to alter the speed of water infiltration during rehydration. We therefore used these infiltration treatments to investigate the effect of different rehydration regimes on DNA damage in FD sperm, fertilization rates following ICSI using FD sperm, embryo development rates, and offspring rates.

## Results

### Simple measurement of solution infiltration speed

Since our laboratory is unable to measure the infiltration speed of solutions into sperm, we used filter paper (Merck Millipore Low Extractable Filters GSWP09000) to provide simple measurements of infiltration speed. Although filter paper and sperm have completely different structures, and it is not possible to predict the infiltration speed into sperm using filter paper, this experiment was conducted to show that the solutions used in this study exhibit different infiltration speeds due to differences in osmolarity, viscosity and temperature. We added droplets of HTF medium, PVP solutions at varying concentrations, and ultrapure water at different temperatures onto filter paper and recorded the diffusion distance per unit time. HTF medium spread more slowly than ultrapure water, PVP solutions showed concentration-dependent diffusion that was consistently slower than water, and ultrapure water spread faster as temperature increased (S1 Table).

### Effects of the osmotic pressure of rehydration solutions on FD sperm

To investigate whether water infiltration speed during rehydration of FD sperm caused DNA damage, we rehydrated FD sperm with either ultrapure water or HTF medium, i.e., the basal medium used for FD sperm production (Fig 1, panels a and b; Fig 2, panels a and b). Next, to assess DNA damage in FD sperm, we performed a comet assay immediately after rehydration with either ultrapure water or HTF medium (Fig 1, panel c). When rehydration was performed with HTF medium, the comet tail, representing DNA damage, showed a slight but significant increase compared to ultrapure water (ultrapure water: 1.00 vs. HTF medium: 1.02, $p < 0.05$) (Fig 2, panels c and d; S2 Table). Next, we injected rehydrated sperm into oocytes via ICSI, and evaluated the fertilization rate, the development rate to the two-cell stage, and the birth rate. We found no significant differences at any of these stages between embryos derived from FD sperm rehydrated with ultrapure water and those rehydrated with HTF medium (Fig 2, panel e; S3 Table). Therefore, our results indicate that rehydration with HTF medium, in which the infiltration speed was slightly slower than in ultrapure water, did not result in reduced sperm DNA damage.

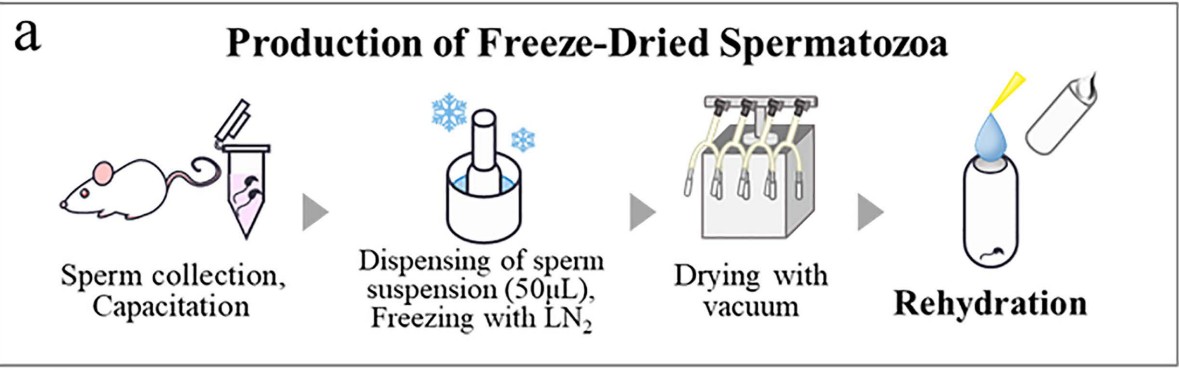

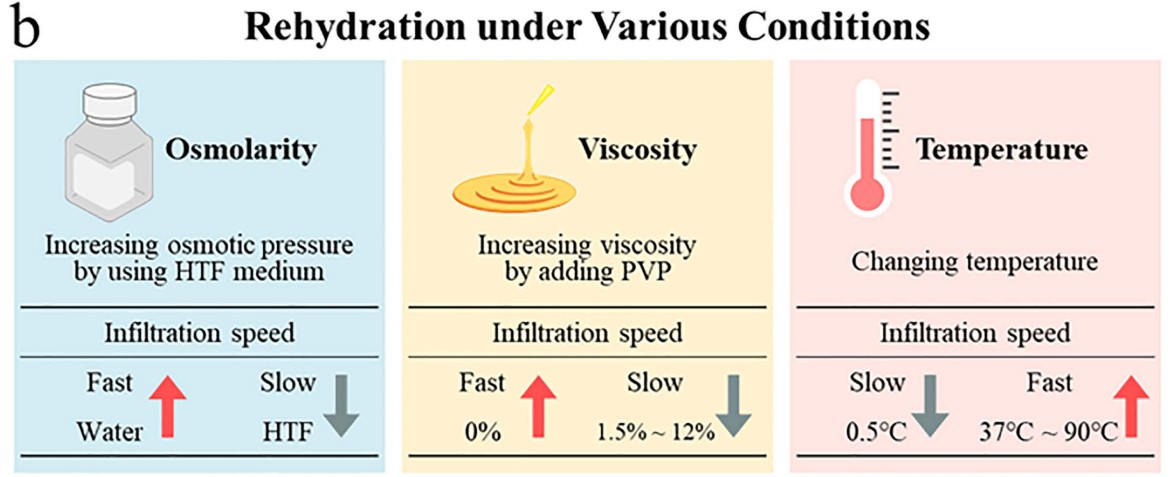

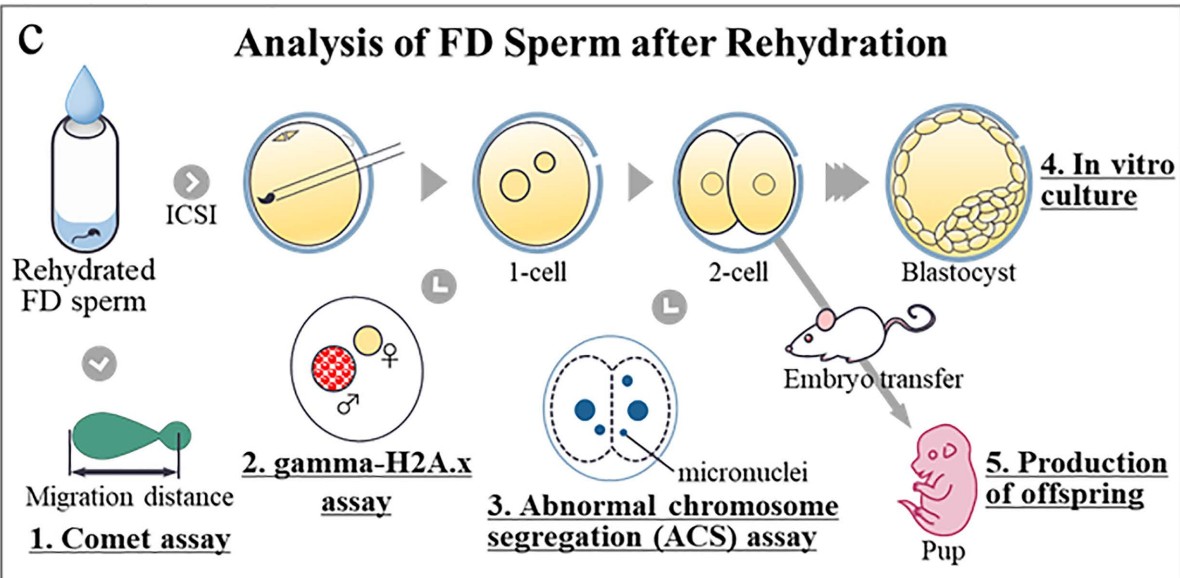

**Fig 1. Schematic representation of preparation of freeze-dried spermatozoa, rehydration conditions, and subsequent experiments. (a)** Spermatozoa were first collected from male mice and were capacitated before being aliquoted into glass ampoules in aliquots of 50 µL. Ampoules were frozen in liquid nitrogen and freeze-dried using a vacuum freeze-dryer. Glass ampoules were then opened prior to subsequent experiments and rehydration was then performed. **(b)** Rehydration was conducted using media subjected to various conditions. To decrease infiltration speed, we used HTF medium with a higher osmotic pressure than ultrapure water. To further reduce infiltration speed, we also used PVP solutions with higher viscosity. In addition,

low-temperature ultrapure water was used to decrease infiltration speed, while high-temperature ultrapure water was used to increase it. **(c)** After adding different rehydration solutions to glass ampoules, we conducted the following experiments: (1) a comet assay to evaluate the degree of DNA damage in spermatozoa, (2) an analysis of gamma-H2Ax foci to assess damage at the pronuclear stage after fertilization, (3) measurements of ACS rates in two-cell embryos, (4) an evaluation of the rate of development to the blastocyst stage, and (5) an assessment of offspring rates after transferring two-cell embryos.

## Effect of rehydration solution viscosity on FD sperm

Next, to investigate whether changes in the infiltration speed caused by the viscosity of the rehydration solution affected the degree of DNA damage, we added PVP to ultrapure water to achieve concentrations ranging from 1.5% to 12%, then used these higher-viscosity solutions for rehydration (Fig 1, panel b). First, we performed a comet assay to evaluate the degree of DNA damage in FD sperm rehydrated with different PVP solutions. Overall, except for the 1.5% and 3% concentrations, we found significant increases in DNA damage as the PVP concentration increased (Fig 2, panels c and d; S4 Table). Since the 6% and 12% PVP solutions caused the most pronounced reduction in infiltration speed (S1 Table) and the greatest increase in DNA damage (Fig 2, panel d; S4 Table), we subsequently used FD sperm rehydrated in the 6% and 12% PVP solutions for ICSI, and we examined ACS at the two-cell stage. ACS can detect large-scale chromosomal damage that cannot be assessed by comet assays. These results showed that ACS rates increased when rehydration was performed using 6% PVP, and the incidence of lethal ACS was 3.5 times higher relative to the ultrapure water control (7% vs. 25%) (Fig 3, panels a and b; S5 Table). Furthermore, in 12% PVP samples, we also observed elevated ACS rates, although not as markedly as for 6% PVP samples. When ICSI was performed using FD sperm rehydrated with 0–12% PVP solutions, we did not observe significant differences in fertilization rates or developmental rates to the two-cell stage. However, we did observe that *in vitro* development rates decreased as the PVP concentration increased, and the blastocyst formation rate was significantly lower for samples containing the highest viscosity solution, 12% PVP (Fig 3, panels c and d; S6 Table). Furthermore, when some of the two-cell stage embryos were transferred to recipient mice to assess birth rates, we also observed a significant decrease in birth rates as the PVP concentration increased (Fig 2, panel e, S7 Table). Based on these results, we concluded that ultrapure water is the most suitable medium for rehydrating FD sperm. In addition, since slower infiltration speeds cause more damage to FD sperm and reduce the birth rate to a greater degree, we next performed experiments using heated ultrapure water to increase infiltration speed during the rehydration of FD sperm (Fig 1, panel b).

## Effect of rehydration solution temperature on FD sperm

To investigate the effect of the water temperature on DNA damage in sperm, we then performed a comet assay on FD sperm rehydrated using medium at each temperature. When rehydration was performed at 0.5°C, we observed significantly higher DNA damage in sperm relative to the RT (23°C–25°C) (Fig 4, panel a; S8 Table). Furthermore, DNA damage significantly decreased as the temperature of the rehydration solution increased above RT, with the lowest DNA damage observed at 70°C (relative value: 1.00 vs. 0.85, $p < 0.05$) (Fig 4, panel a; S8 Table).

Next, we performed ICSI using FD sperm rehydrated at specific temperatures. In preliminary experiments, when rehydrated with water kept at temperatures between 0.5°C to 50°C, the oocyte activation rate following ICSI did not decrease. However, when rehydrated with water at 70°C or 90°C, some oocytes did not activate after ICSI. Therefore, the 70°C or 90°C rehydration oocytes were artificially activated with $SrCl_2$ after ICSI (Table 1). When male pronuclei of one-cell embryos were examined using gamma-H2Ax antibodies, we found that rehydration at 0.5°C resulted in higher gamma-H2Ax fluorescence intensity relative to RT rehydration, which indicates that increased DNA damage was present (Fig 4, panel b and c; S9 Table). In contrast, rehydration at 37°C resulted in the lowest gamma-H2Ax fluorescence intensity compared to the RT control, which is suggestive of reduced DNA damage. However, rehydration at temperatures above 50°C showed DNA damage levels that were comparable to or slightly higher than those observed in samples dehydrated at RT.

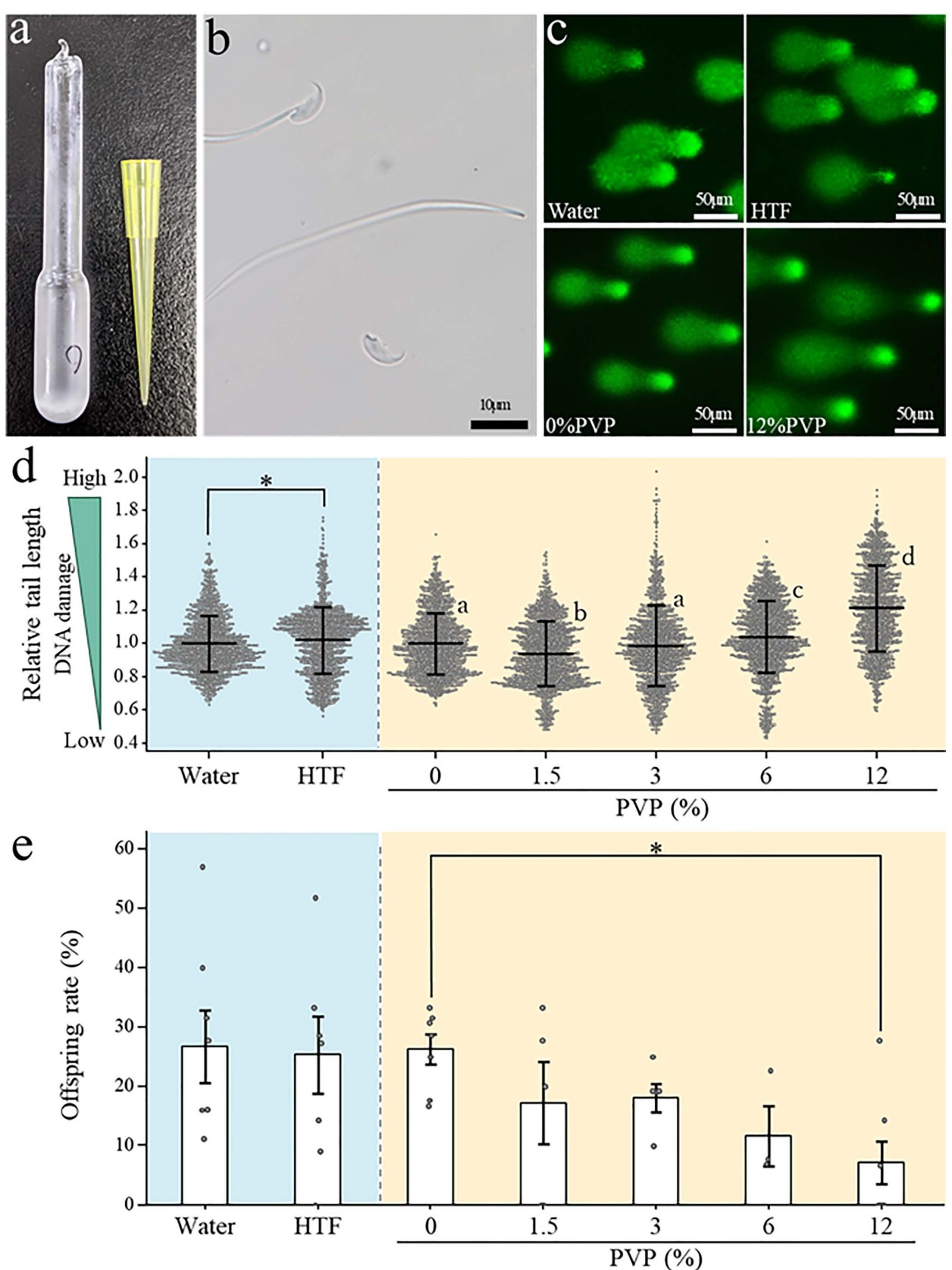

**Fig 2. DNA damage, and offspring rates under rehydration conditions under varying osmotic pressure or viscosity. (a)** Glass ampoules containing freeze-dried (FD) spermatozoa. **(b)** Freeze-dried spermatozoa showing separation of the sperm head during freeze-drying. **(c)** Comet tail length of FD sperm was compared between ultrapure water control group (top left), the HTF medium experimental group (top right), the 0% PVP solution control group (bottom left), and the 12% PVP solution experimental group (bottom right). **(d)** DNA damage in FD sperm rehydrated with HTF medium or PVP solution evaluated using comet assays. Background colors on the graph represent different conditions; blue indicates experimental conditions where osmotic pressure was altered using HTF medium, yellow represents conditions where viscosity was modified using PVP solution. Comet tail lengths were normalized to those of FD sperm rehydrated with ultrapure water (control). The vertical axis represents relative DNA damage, with higher values indicating greater damage. Each data point represents a value recorded for an individual spermatozoon, with error bars showing standard deviation. An asterisk denotes statistically significant differences between pairs of samples ($p < 0.05$). Different letters (a vs. b, c, d; b vs. c, d; c vs. d) indicate statistically significantly different group means ($p < 0.05$). **(e)** Offspring rates were evaluated in intracytoplasmic sperm injection (ICSI) embryos using FD sperm rehydrated with HTF medium or PVP solution. Background colors on the graph represent different conditions; blue indicates experimental conditions where osmotic pressure was altered using HTF medium, yellow represents conditions where viscosity was modified using PVP solution. Asterisks denote statistically significant differences between pairs of samples ($p < 0.05$).

Next, we examined ACS in two-cell embryos and found that rehydration at 0.5°C increased the incidence of severe ACS (i.e., classified as "heavy" or "lethal") relative to the RT control (Fig 4, panels d and e; S10 Table). In contrast, increasing the rehydration temperature above RT tended to reduce the incidence of severe ACS, with the least damage observed at 50°C. Moreover, embryos showing light ACS can sometimes result in viable offspring [31]; if we combine both those embryos scored as normal chromosome segregation (NCS) and Light as "normal" embryos, and the remaining embryos to be "abnormal" embryos, we find that the lowest proportion of abnormal embryos (29%) was observed at 50°C. Finally, we transferred two-cell embryos to recipients and evaluated the resulting birth rate. We found that rehydration at 0.5°C resulted in a birth rate of 23%, which was similar to the rate obtained with RT (22%) (Table 1). However, rehydration at temperatures above RT improved the birth rate to 29% (at 37°C) and 37% (at 50°C) (Fig 4, panels f and g). In contrast, rehydration at 70°C or higher resulted in a slight decrease in birth rate compared to 50°C (70°C: 31%; 90°C: 35%).

## Discussion

Previous studies have reported that several kinds of damage affect sperm during the drying process. However, this study revealed—for the first time—that sperm DNA can also be damaged during rehydration. In addition, here we show that rehydrating with high-temperature ultrapure water—which has previously been considered to be harmful to sperm—reduces DNA damage in FD sperm. Furthermore, using conventional methods, the birth rate for mouse FD sperm-derived embryos is approximately 20%, but when following a rehydration protocol using ultrapure water at 50°C, we observed birth rates as high as 37%.

Here, we first hypothesized that—just as riverbanks erode when exposed to rapid currents [32]—faster infiltration speeds may damage the FD sperm. Previous research on various organisms and organelles, such as microorganisms, seeds, and liposomes, has shown that rehydration conditions influence recovery from a dried state [33–37]. For example, rapid rehydration may cause osmotic shock and/or sudden membrane phase transitions, both of which can induce rehydration-specific damage, including membrane rupture, organelle degeneration, and oxidative stress. We therefore predicted that our rehydration protocol could cause severe FD sperm damage. However, because FD sperm are directly injected into oocytes by ICSI, membrane or mitochondrial damage may not affect fertilization or embryo development. Furthermore, rapid rehydration may influence the DNA or protamine content of FD sperm nuclei. Based on this hypothesis, we proposed that employing a slower infiltration method during rehydration of FD sperm can mitigate DNA damage and improve the birth rate. However, despite attempts to slow down the infiltration speed by increasing osmotic pressure and viscosity, we found—contrary to our expectations—that lower infiltration speed was associated with increased DNA damage in FD sperm, as well as significant decreases in blastocyst formation and birth rates. In contrast to our initial hypothesis, when pure water was heated to high temperatures to speed up infiltration into the nucleus, we observed reduced DNA damage in FD sperm, as well as birth rates that were up to 1.8 times higher than those in FD sperm rehydrated with

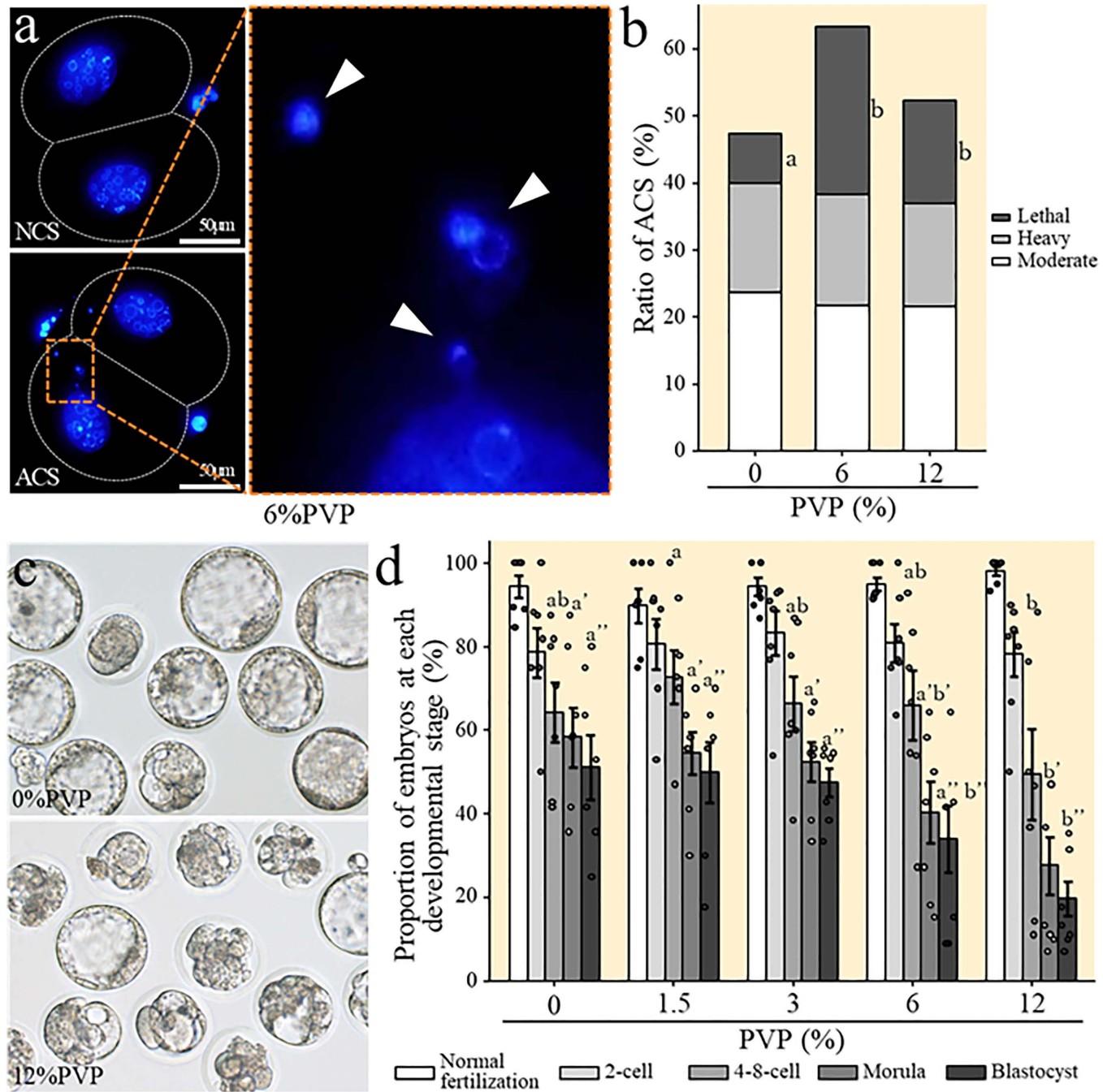

**Fig 3. ACS level and rates of development to the blastocyst stage for FD sperm were rehydrated at different PVP concentrations. (a)** Two-cell embryos derived from FD sperm rehydrated with 6% PVP and stained with DAPI. Upper left image shows a two-cell embryo with normal chromosome segregation (NCS), while the lower left image shows a two-cell embryo with abnormal chromosome segregation (ACS). The enlarged image on the right shows a highlighted portion of the ACS embryo, and the arrowhead indicates the micronucleus. **(b)** Proportion of embryos with moderate or higher ACS among all embryos derived from FD sperm rehydrated with different PVP concentrations. Different letters indicate statistically significantly different group means ($p<0.05$). **(c)** Blastocyst images. The top panel shows blastocysts derived from embryos originating from FD sperm rehydrated with 0% PVP (i.e., control). The bottom panel shows blastocysts derived from embryos originating from FD sperm rehydrated with 12% PVP (i.e., experimental group). **(d)** Rate of development to the blastocyst stage for embryos derived from FD sperm rehydrated under different PVP concentrations. Each data point represents an independent measurement of the proportion of embryos reaching each stage relative to normally fertilized embryos. Bars indicate the mean and error bars represent the standard error of the mean (SEM).

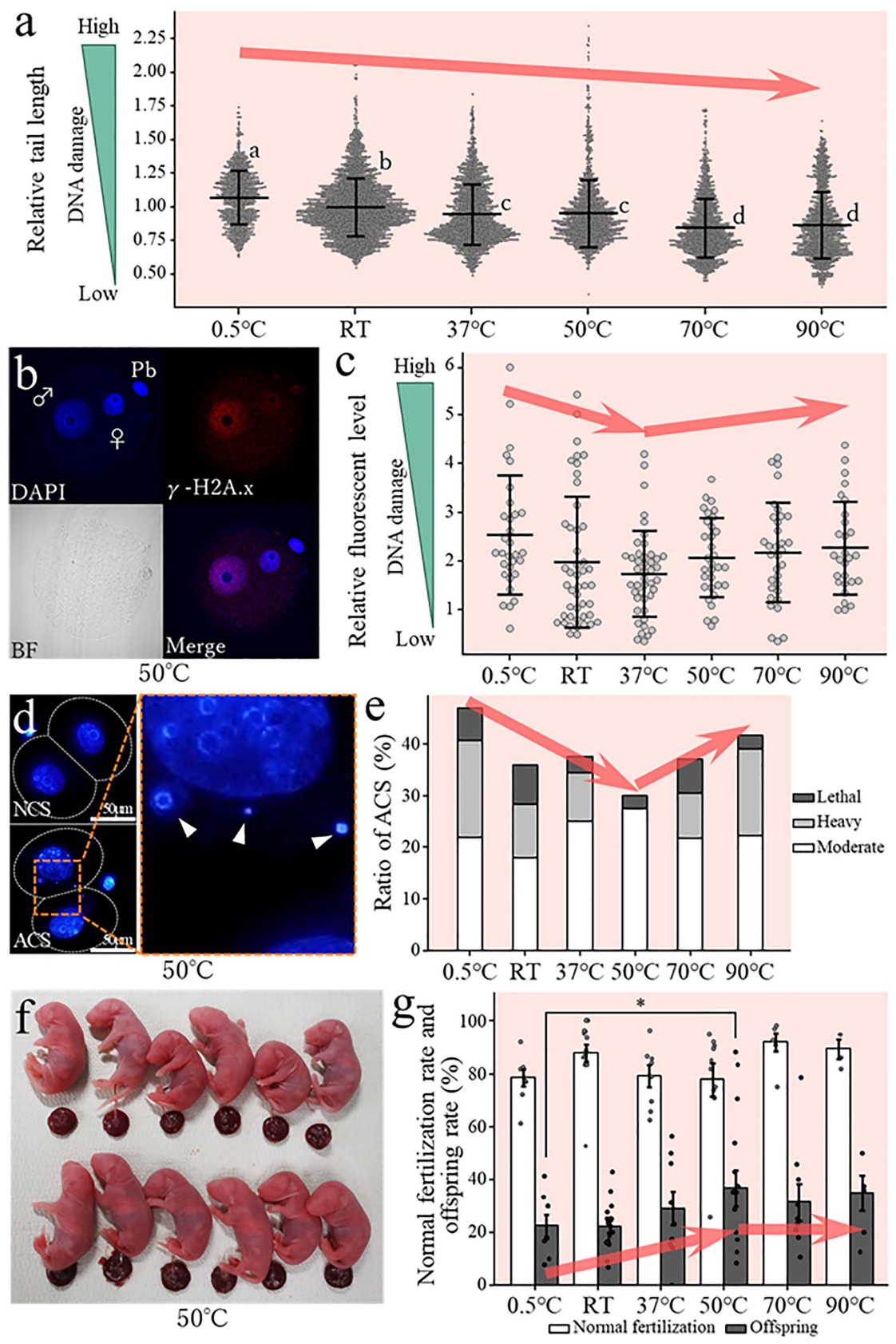

**Fig 4. Changes in DNA damage and developmental potential following rehydration using ultrapure water at different temperatures. (a)** Observed DNA damage in FD sperm rehydrated with ultrapure water at different temperatures as evaluated using comet assays. Different letters indicate statistically significantly different group means ($p < 0.05$). **(b)** gamma-H2Ax assay of fertilized embryos derived from FD sperm rehydrated with ultrapure water at different temperatures. Images show male and female pronuclei stained with 4′6-diamidino-2-phenylindole (DAPI, blue, top left), gamma-H2Ax signals indicating double-stranded DNA breaks (red, top right), bright-field images (bottom left), and merged images (bottom right). **(c)** Relative brightness of male pronuclei sourced from fertilized embryos derived from FD sperm rehydrated at different temperatures. Higher brightness values indicate more DNA damage. **(d)** Two-cell embryos derived from FD sperm rehydrated with ultrapure water at 50°C after staining with DAPI. The upper left image shows a two-cell embryo with normal chromosome segregation (NCS), while the lower left image shows a two-cell embryo with abnormal chromosome segregation (ACS). The enlarged image on the right shows a highlighted portion of the ACS embryo, and the arrowhead indicates the micronucleus. **(e)** Proportion of embryos with moderate or higher ACS among all embryos derived from FD sperm rehydrated with ultrapure water at different temperatures. **(f)** Offspring derived from ICSI embryos using FD sperm rehydrated with ultrapure water at 50°C. **(g)** Normal fertilization rates and offspring rates of ICSI embryos using FD sperm rehydrated with ultrapure water at different temperatures. Asterisks denote statistically significant differences in sample means ($p < 0.05$).

pure water at RT. Taken together, these results suggest that during FD sperm rehydration, rapid water infiltration into the nucleus is more effective in preventing DNA damage than slower water infiltration.

It has been reported that when cellular DNA experiences drying, it adopts a compact A-DNA structure [38]. In sperm, the nucleus is composed of protamine instead of histones [39], making it unclear whether drying can induce an A-DNA structure in sperm DNA as it does in other cell types. As FD treatment alters the sperm plasma membrane and can cause DNA damage [7], structural changes to DNA are likely. Once dried sperm are rehydrated, inconsistencies may occur between surface DNA regions, which recover quickly, and central regions, which remain dry during early water infiltration. Such inconsistencies may lead to chromosomal damage and exacerbate minor DNA lesions caused by FD processing. Therefore, when rehydration is performed with hot water, which infiltrates the nucleus quickly, the structure of DNA may be restored uniformly and immediately, thereby minimizing structural inconsistencies and consequently improving the birth rate. Indeed, when we evaluated the DNA damage on each FD sperm treatment, we found that rehydration at low temperatures increased DNA damage relative to the RT control. Conversely, rehydration at higher temperatures resulted in reduced DNA damage.

However, from comet assay and ACS assay [40], although reducing minor and severe DNA damage as the temperature increases, severe DNA damage can increase when temperatures exceed over 50°C. In the past, heat treatments of fresh sperm were considered undesirable since they were known to impair the oocyte activation potential of spermatozoa [41,42]. When fresh sperm are subjected to heat treatment, complete loss of oocyte activation occurs at 56°C, and no offspring can be obtained at 80°C even with artificial activation of oocytes following ICSI. In this study, the maximum temperature of the hot ultrapure water was 90°C, and only 50 μL was added to the FD sperm sample. Although only 50 μL of hot ultrapure water was added, which likely caused the temperature to drop rapidly during rehydration, rehydrated sperm under such conditions may have become as heat-sensitive as suggested by the reduced oocyte activation potential rate observed with FD sperm rehydrated at 70°C and 90°C. However, even if FD sperm lost activation potential due to hot water rehydration, offspring may still be obtained by performing artificial activation of oocytes. Similarly, FD sperm stored at room temperature for 6 years or briefly exposed to temperatures as high as 150°C, exhibit reduced activation capacity. However, live offspring can be obtained by performing artificial activation of oocytes after ICSI. Thus, it is likely that FD treatment enhances DNA resistance rather than oocyte activation factors on sperm.

Next, we observed that birth rates improved as the temperature of the ultrapure water increased, with the highest birth rates being recorded at 50°C. Interestingly, we observed higher levels of severe DNA damage in sperm when water was added at over 50°C, but the birth rate did not decrease. Perhaps when hot water at temperatures over 50°C is added to FD sperm, the number of embryos with chromosomal abnormalities increases, thereby reducing the proportion of embryos that can develop into live offspring. However, as the water temperature increased, minor DNA damage continued to decrease, leading to an improvement in the overall quality of the embryos. As a result, the rate of live offspring remained

**Table 1. Full-term development rates of ICSI embryos using FD sperm rehydrated with ultrapure water at different temperatures.**

| Temperature (°C) | Oocyte activation | No. of oocytes surviving after ICSI [no. of ICSI procedures] | No. (%) of oocytes surviving after oocyte activation | No. (%) of fertilized zygotes | No. (%) of embryos developed to 2 cell | No. of transferred embryos [no. of recipients] | No. (%) [min-max] of offspring |
|---|---|---|---|---|---|---|---|
| 0.5 | - | 186 [8] | - | 146 (78) | 122 (85) | 124 [8] | 28 (23)[ab] [8-41] |
| RT | - | 333 [14] | - | 293 (88) | 251 (86) | 251 [16] | 56 (22)[a] [7-43] |
| 37 | - | 202 [8] | - | 160 (79) | 132 (83) | 117 [9] | 34 (29)[ab] [0-56] |
| 50 | - | 333 [10] | - | 259 (78) | 237 (92) | 226 [14] | 83 (37)[b] [8-88] |
| 70 | - | 44 [3] | - | 33 (75) | 28 (85) | 28 [2] | 4 (14)[ab] [0-17] |
| 70 | + | 282 [6] | 251 (89) | 231 (92) | 184 (80) | 175 [9] | 55 (31)[ab] [11-79] |
| 90 | - | 59 [3] | - | 48 (81) | 42 (88) | 23 [3] | 8 (35)[ab] [13-60] |
| 90 | + | 172 [3] | 141 (82) | 126 (89) | 85 (67) | 83 [5] | 29 (35)[ab] [13-50] |

unchanged. In addition, if it were possible to exclude ACS embryos before embryo transfer, or to reduce the incidence of ACS [26], we speculate that the birth rate might be further improved.

Overall, this study revealed that DNA damage occurs not only during FD but also during rehydration, and that rapid infiltration can suppress this damage. However, the birth rate of embryos derived from FD sperm remains lower than that of fresh sperm. To further improve birth rate, it is necessary to elucidate which mechanisms of DNA damage are induced by all processes involved in FD sperm production and to develop new methods for minimizing this damage. If DNA damage can be prevented throughout the process, it should be possible to achieve birth rates comparable to those of fresh sperm. In recent years, global warming and the spread of emerging infectious diseases has highlighted the importance of genetic resources [1–3]. While sperm is the most suitable genetic resource, for individuals from whom sperm cannot be collected, a round spermatid (i.e., a sperm progenitor cell) can be harvested from infertile males [43,44] and somatic cells [45], which are present in any individual regardless of age or sex, are also a valuable genetic resources [46]. Although research on FD these cells remains ongoing, the success rate remains very low, and a practical use has not yet been established [47,48]. If rapid infiltration using hot water can reduce DNA damage and improve the birth rate of FD cells, their utility as genetic resources would increase significantly. Although oocytes have not yet been successfully preserved using FD techniques, the chances of success may increase if they are rehydrated using hot water. The findings of this study are expected to contribute significantly to future enhancements of the reliability and practical application of FD technology for genetic resource preservation.

## Materials and methods

### Animals

Female and male ICR mice (8–10 weeks old) were first obtained from SLC Inc. (Hamamatsu, Japan). Surrogate pseudo pregnant ICR females were then prepared for embryo implantation by mating with vasectomized ICR males that had been previously confirmed as sterile. All mice were euthanized (see Embryo transfer of this Method section) either on the day

on which the experiment was performed or following the completion of experiments via $CO_2$ inhalation or cervical dislocation. All animal experiments followed the ARRIVE animal care guidelines. Moreover, all experiments were also performed in accordance with the guidelines for the Care and Use of Laboratory Animals and the specific experimental protocols employed in this study were approved by the Institutional Committee of Laboratory Animal Experimentation of the University of Yamanashi (Ref. No.: A29-24).

## Media

HTF medium [49] was used for capacitation and FD of spermatozoa as well as for spermatozoa rehydration. HEPES-CZB [50] and CZB [51] media were used for oocyte/embryo manipulation and embryos were incubated in 5% $CO_2$ at 37°C.

## Preparation of FD spermatozoa

To prepare FD sperm, both epididymides were first collected from male ICR mice, and the ducts were severed using sharp scissors. A few drops of the dense spermatozoa mass were then added into a centrifuge tube containing 850 µL of HTF medium. This mixture was then incubated for 30 min at 37°C and 5% $CO_2$. In this study, to examine the effect of rehydration on FD sperm, we decided to produce FD sperm in an HTF medium that did not contain freeze-drying protectants. Next, the concentration and motility of spermatozoa were determined under a microscope, and 400 µl of supernatant was collected. An aliquot of 50 µl spermatozoa suspension was then dispensed into each glass ampoule. Next, the ampoules were flash frozen in $LN_2$ then freeze-dried using an FDU-2200 freeze dryer (EYELA, Tokyo, Japan) [12,15]. The cork of the freeze-dryer was opened for at least 6 h until all samples had completely dried. After drying, ampoules were sealed by melting their glass necks under vacuum using a gas burner. All ampoules were then stored in a freezer at −30°C to avoid damage during storage and used within one month. In this study we did not analyze the effect of storage duration in detail. However, previous studies have demonstrated that room temperature storage for up to six years does not adversely affect sperm DNA integrity or fertility. Therefore, we considered the impact of the short-term storage (i.e., under −30°C for up to one month) performed here to be negligible.

## Detection of air trapped in ampoules using a Tesla coil leak detector

Air within the ampoules were detected using a Tesla coil leak detector (Sanko Electronic Laboratory, Kanagawa, Japan) with all procedures performed according to the manufacturer's instructions. When the tip of the Tesla coil is brought near the ampoule, the tip forms sparks near the glass. However, if a large amount of air is trapped within the ampoule, this air cannot be ionized. However, if the ampoule contains only a small amount of residual air, its ionization produces a spark within the ampoule. Only Tesla-positive ampoules, i.e., those containing almost no air if not no air whatsoever, were used for all experiments [12,13].

## Rehydration of FD spermatozoa

For rehydration, we first opened FD sperm ampoules immediately prior to use, and to each ampoule 50 µL of rehydration solution was added. The rehydration solution used was specific to each experimental condition. After this addition, ampoule contents were immediately mixed thoroughly by pipetting multiple times. Room temperature ultrapure water (i.e., Milli-Q purified, resistivity 18.2 MΩ·cm) was used as the control solution for all experiments. Ultrapure water has low osmotic pressure and is not suitable as a cell culture medium. Here, sperm was freeze-dried in the culture medium; therefore, adding the same volume of water will restore the environment to one suitable for sperm after rehydration. For osmotic pressure experiments, we used HTF medium (Fig 1, panel b (left)), while for high-viscosity experiments, we used PVP solutions with concentrations of 1.5%, 3%, 6%, and 12% (Fig 1, panel b (center)). Finally, for temperature experiments, we used ultrapure water heated to 0.5°C, 37°C, 50°C, 70°C, or 90°C (Fig 1, panel b (right)). The temperature of ultrapure water was adjusted using an ice-water bath or a water bath.

## Oocyte preparation

To superovulate female mice, we injected each with 5 IU of equine chorionic gonadotropin, followed by 5 IU of human chorionic gonadotropin 48 h later. After 14–16 hours, we collected cumulus-oocyte complexes (COCs) from the oviducts of all female mice and transferred the COCs to a Falcon dish containing HEPES-CZB medium. Next, we dispersed the cumulus by transferring COCs to a 50 μl droplet of HEPES-CZB medium containing 0.1% bovine testicular hyaluronidase and incubating at RT for 3 minutes. The resulting cumulus-free oocytes were then washed twice before being transferred to a 20 μL droplet of CZB for further culturing.

## ICSI and artificial oocyte activation

ICSI was performed as per a previously described protocol [50], and the sperm involved were prepared as mentioned above. Ampoules from each experimental group were opened, rehydrated, and ICSI was performed immediately. Therefore, the elapsed time between the start of rehydration and ICSI was constant for all experiments. For each sperm microinjection, 1–2 μL of sperm suspension was transferred directly to the injection chamber without washing. The sperm suspension was then replaced every 30 min during ICSI. For injection, the sperm head was first separated from the tail by applying piezoelectric pulses, and the tailless head was then microinjected into the oocyte. Next, all oocytes that survived ICSI were incubated in CZB medium at 37°C with 5% $CO_2$. After 15 min, some oocytes were artificially activated by immersion in 5 mM strontium chloride for 1 h. After incubation, these were cultured again at 37°C in 5% $CO_2$ CZB medium. Finally, pronucleus formation was assessed 6 h after ICSI.

## Embryo transfer

For embryo transfer, embryos at the two-cell stage implanted into a day 0.5 pseudo pregnant ICR female mouse that had mated with a vasectomized male the night before the transfer. On the day of embryo transfer, recipients were first anesthetized via intraperitoneal injection of medetomidine, midazolam, and butorphanol. After completion of embryo transfer, atipamezole was administered, and mice were kept warm until they regained consciousness. A total of 6–10 embryos were transferred into each oviduct. On day 18.5 of gestation, offspring were delivered via cesarean section and allowed to mature normally. Remaining unused embryos were cultured for up to four days to evaluate their potential for development into blastocysts.

## Analysis and scoring of comet slides

DNA damage to spermatozoa, including single- and double-strand breaks [52] was measured using a CometAssay™ Kit (Trevigen, MD, USA) with all procedures performed as per the manufacturer's protocol. Briefly, spermatozoa specimens were first collected from ampoules immediately after opening and were then rehydrated in water. Specimens were then mounted on slides without washing, and 100–300 spermatozoa heads on each slide were subsequently analyzed via electrophoresis. To standardize the results obtained from the different conditions under which spermatozoa were produced, the length of each DNA comet tail was divided by the mean length of the one-sided results of each experiment.

## Gamma-H2Ax assays

Histone H2Ax is an important variant of H2A, as it contains a serine residue at position 139 that is rapidly phosphorylated within seconds of DNA damage. The resulting phosphorylated H2Ax, known as gamma-H2Ax, then forms foci at the sites of DNA damage, which leads to the recruitment of various repair and cell-cycle checkpoint proteins. Given this role, we used gamma-H2Ax foci formation as a marker of DNA double-strand breaks observed in male and female pronuclei.

   For this experiment, all specimens were first fixed with 4% paraformaldehyde (PFA; Wako Pure Chemical, Osaka, Japan) containing 0.2% Triton X at RT for 20 min before being stored in a refrigerator until staining. The primary antibody

used for zygote immunostaining was the anti-phospho-H2Ax (Ser139) rabbit polyclonal antibody (1:500; Millipore-Merck, Darmstadt, Germany), while the secondary antibody used was the Alexa Fluor 568-labeled goat antirabbit IgG (1:500; Molecular Probes). Moreover, DNA was stained using 4′6-diamidino-2-phenylindole (2 µg/ml; Molecular Probes). We detected several DNA repair sites in male pronuclei. However, we found that counting the number of repair sites within a pronucleus was challenging. Therefore, the brightness of each male pronucleus was measured using NIH ImageJ and was then subtracted from the brightness of the zygote cytoplasm.

### Detection of abnormal chromosome segregation (ACS)

One day after ICSI, embryos at the two-cell stage were fixed and permeabilized with 4% PFA and 0.5% Triton X-100 for 15 min. These embryos were then immersed in PBS containing DAPI and 1% BSA before being analyzed using a fluorescence microscope (Olympus IX-73, Tokyo, Japan). As previously described, we classified ACS as belonging to on of four groups: light, moderate, heavy, and lethal. Specifically, light ACS was defined as the presence of only one micronucleus [19], moderate ACS by the presence of two small micronuclei, one to two medium micronuclei, or a single large micronucleus. Heavy ACS involved detection of three small or medium micronuclei or two to three large micronuclei, while lethal ACS involved embryos that contained multiple micronuclei. In some instances, two conditions occurred simultaneously, leading to a more severe evaluation. For example, if an embryo contained one medium and two small micronuclei, we classified it as "heavy." However, since embryos with low-level ACS can develop normally—i.e., to full term [31]—we instead focused on embryos with severe (i.e., moderate or higher) ACS in this study.

### Statistical analysis

Comet assay results were analyzed using Wilcoxon–Mann–Whitney nonparametric tests or Dunn's tests. Gamma-H2Ax assay results were analyzed using Kruskal–Wallis tests. Normal fertilization rates, ACS assays, *in vitro* development rates, and birth rates were evaluated using chi-square tests or Tukey's WSD tests. For all statistical tests, statistical significance was determined as $p < 0.05$.

### Supporting information

**S1 Table. Simplified measurement of infiltration speeds of rehydration solutions under different conditions.**
(TIF)

**S2 Table. Comet assay results for FD sperm rehydrated with media of varying osmotic pressure.**
(TIF)

**S3 Table. Full-term developmental rates of ICSI embryos derived from FD sperm rehydrated with media of varying osmotic pressures or different PVP concentrations.**
(TIF)

**S4 Table. Comet assay of FD sperm rehydrated with PVP solutions at different concentrations.**
(TIF)

**S5 Table. Abnormal chromosome segregation (ACS) rates in two-cell embryos derived from FD sperm rehydrated with PVP solutions at different concentrations.**
(TIF)

**S6 Table. Developmental rates to the blastocyst stage of ICSI embryos derived from FD sperm rehydrated with PVP solutions at different concentrations.**
(TIF)

**S7 Table. Full-term development rates of ICSI embryos using FD sperm rehydrated with solutions of different PVP concentrations.**
(TIF)

**S8 Table. Comet assay of FD sperm rehydrated with ultrapure water at different temperatures.**
(TIF)

**S9 Table. Brightness of male pronuclei derived from FD sperm rehydrated with ultrapure water at different temperatures, immunostained with the anti-gamma-H2Ax antibody.**
(TIF)

**S10 Table. Abnormal chromosome segregation (ACS) rates in two-cell stage embryos derived from FD sperm rehydrated with ultrapure water at different temperatures.**
(TIF)

## Acknowledgments

We thank Y. Kanda for assistance in preparing this manuscript.

## Author contributions

**Formal analysis:** Teruhiko Wakayama.

**Funding acquisition:** Teruhiko Wakayama.

**Investigation:** Kango Yamaji, Sayaka Wakayama, Natsuki Ushigome, Daiyu Ito.

**Writing – original draft:** Teruhiko Wakayama, Kango Yamaji.

**Writing – review & editing:** Teruhiko Wakayama.

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
