## [Decision Letter · Decision Letter 0]

28 Jul 2025

Dear Dr. Wakayama,

Major comment:

Overall, the discussion on the potential mechanism of the surprising finding is somewhat shallow. Would cellular organelle structures be affected by faster rehydration? Are there studies of rehydrating other materials, microorganisms or even plants, that can provide some clues on the mechanisms?

Minor comments:

Brackets of references: some are separated by a space from the preceding words, others are not. Please look at the Journals requirements and be consistent.  

L58: change “F” to “f”.

L69: “FD pretreatment” is not clear.  Is FD the pretreatment?  If so, the word “pretreatment” is not needed. If not, please give an example of the pretreatment using “such as”.

L72: “over 200 years”, is this a mistake? I don’t believe that the International Space Station is that old.

L76: “one-third” of “fresh or cryopreserved sperm”. This implies that fresh and cryopreserved sperm have the same birth rates. Is this correct?  If not, please provide two fractions.

L80: please insert the reference of your prior publication.

L90: “first sperm freezing”? It seems “freezing” alone is more appropriate.

L89-98: this section confuses the readers. First FD is described as a 4-step procedure, but then it was stated that drying without freezing can also be done.  These are conflicting statements. Please revise.

L126: please give some details of the filter paper because not all filter papers are the same.

L256: “structural changes”, it is unclear what structure is being discussed here, plasma membrane, organelles or DNA? The following statements also need to be clearer in the specific structures being discussed.  

Additionally, please also address comments from Reviewer 1.

We look forward to receiving your revised manuscript.

Kind regards,

Xiuchun Tian

Academic Editor

PLOS ONE

 [This work was partially funded by JST SPRING, Grant Number JPMJSP2133 to N. U.; the Research Fellowships of Japan Society for the Promotion of Science for Young Scientists to D.I. (23K19330), S. W. (23K08843) to T.W. (23K18124 and 24K01779); the Naito Foundation and Takahashi Industrial and Economic Research Foundation (189) to S.W.; Asada Science Foundation and the Canon Foundation (M20-0008) to T.W.]. 

Additional Editor Comments (if provided):

Reviewers' comments:

Reviewer's Responses to Questions

**Comments to the Author**

1. Is the manuscript technically sound, and do the data support the conclusions?

Reviewer #1: Yes

2. Has the statistical analysis been performed appropriately and rigorously?

Reviewer #1: I Don't Know

3. Have the authors made all data underlying the findings in their manuscript fully available?

Reviewer #1: Yes

4. Is the manuscript presented in an intelligible fashion and written in standard English?

Reviewer #1: Yes

Reviewer #1: PONE-D-25-31910

Improved Birth Rates via Rehydration of Mouse Freeze-Dried Spermatozoa using High-Temperature Ultrapure Water

General comments:

This paper describes effects of different rehydration rates on DNA damage and fertilization potential of freeze-dried (FD) mouse sperm. More specifically, FD sperm were rehydrated using HTF medium instead of water or water supplemented with PVP for decreasing the infiltration speed, while water of increasing temperatures was used for obtaining more rapid rehydration. The comet was employed to assess sperm DNA damage after rehydration. In addition, after fertilization, gamma-H2A.x immunostaining and abnormal chromosome segregation patterns were investigated. ICSI with FD sperm was performed to determine fertilization potential and in vitro development of sperm rehydrated using different protocols, while embryo transfers with obtained 2-cell stages were used to assess if these could lead to producing offspring. It was found that rehydration of FD-sperm with water containing increasing PVP concentrations was damaging. Furthermore, rapid rehydration of FD sperm, i.e., with water of 50C, resulted in decreased sperm DNA damage, which in turn increased ICSI outcomes and birth rates. Rehydration with water of 70 or 90C resulted in lower sperm DNA damage, but resulted in the need of artificial oocyte activation following ICSI.

Major points:

1) This is a full paper, in which multiple aspects were investigated; focusing on effects of rehydration-related damage of FD sperm. Although it may not be unexpected that sperm biomolecular damage progresses upon extended rehydration duration, due to accumulation of reactive oxygen species, and use of high PVP concentrations with ICSI impairs development; it is important to document these data. These authors published several papers on freeze-drying of mouse sperm, and here merely use their established protocols and assays while testing a further variable. Anyways, the authors present a lot of work, with testing multiple concentrations/treatments resulting in dose-dependent effects due to both slower and faster rehydration, while correlating sperm DNA damage with ICSI outcomes as well as offspring when used for embryo transfer.

2) The rationale of several methodologies is not clear. Why was HTF used as a base medium for freeze drying, without use of EDTA, antioxidants and/or disaccharides as has been described beneficial by others? Was the duration between starting rehydration and performing ICSI kept constant, i.e., was ICSI performed after varying durations in case of slow vs rapid rehydration and can differences simply be explained by the time sperm spend in their rehydration medium? Were sperm washed after rehydration, i.e., for avoiding interference of different types and concentrations of compounds with the post-rehydration assays performed (comet assay, ICSI)?

3) It is appreciated that the authors aimed to validate that their different approaches result in different rehydration rates, however, infiltration in a paper strip does not reflect water movement into cellular structures (i.e., containing membranes/organelles and compartments with different compositions). The authors note this. Anyways, describing physical properties of the rehydration solutions used (osmolality, viscosity, temperature) may be sufficient here, and the description in L119-136 can be shortened/presented more condensed (and supplemental figure 1 can be omitted).

4) Providing tables with numbers on actual numbers of sperm/oocytes/embryos studies as supplemental data is good. Supplemental figure 2, however, is maybe better presented in the main document. Why did the authors decide presenting these data in the supplement or can this be moved?

4) The discussion section is relatively long, and possibly can be shortened while focusing on the major findings and omitting repetitions. Also the introduction can possibly presented in a more condensed manner. The reference listing contains relatively many papers from the same authors.

Minor points:

L2: what is ‘ultrapure water’; and is this needed?

L15: maybe rephrase the short title; anyways, write ‘of’ instead of ‘of via’

L24: delete ‘a previously unexamined’, better just write what you did, i.e., investigating the effect of different rehydration rates?

L59: can the authors elaborate on possible differences amongst species, i.e., their susceptibility for FD-related damage, DNA stability in the dried state, and success rates with ICSI?

L61: can the authors elaborate on other factors affecting storage stability, like the storage temperature, sample water content, freeze-drying formulation/protectants used, storage under nitrogen gas, etc; and cite work from others on these matters too?

L69: what is meant with ‘pretreatment’?

L79: example of a repetition; see L62

L84: example of a repetition; see L81

L89-100: this reviewer is not sure if the presented rationale explains that damage with FD sperm predominantly originates from the drying step

L91: what about use of disaccharides like trehalose; which can act both as a cryoprotectant and lyoprotectant?

L100: rephrase, awkward statement: ‘whereas introducing a freezing step mitigates damage caused by drying’

L104-105: rephrase, awkward statement: ‘did not mitigate damage… and was found to significantly improve birth rates’

L113: just a thought. what is the most important factor causing damage: the speed of rehydration, incubation duration after rehydration, or extent of damage accumulated during storage in the dried state?

L119: see comment above (3): this section can possibly shortened

L137: just a thought: what about including fresh/shock-frozen controls (resuspended in similar media) for seeing comet tail lengths and ICSI outcomes with those?

L140: please carefully check if references to the different figures/panels is correct.

L158: see comment above (2): was PVP as present in the rehydration solution removed/were different PVP contents injected with ICSI? are similar effects found when using freshly diluted (or cryopreserved) sperm in solutions containing increasing PVP concentrations; i.e., both when analyzed using the comet assay and when used for ICSI?

L202: this reviewer is intrigued by the phenomenon that DNA damage decreases in a dose-dependent manner when rehydrating FD-sperm with water of increasing temperatures, while artificial activation with ICSI is only needed when using water of 70 or 90C. remember that in their earlier recent paper (Kamada et al 2025; Sci Rep 15:303. doi: 10.1038/s41598-024-83350-2), the authors found that artificial activation was also needed when FD sperm were stored for long durations; i.e., it seems that inactivation/degradation/loss of activation factors needed for oocyte activation following ICSI can be simply compensated for by SrCl2-treatment (as long as the sperm DNA is intact)? it would be good when the authors highlight this finding better, and give their insights in the causes of sperm losing their oocyte activation capacity after exposure to specific conditions.

L229: see comment above (4): this section can possibly be shortened

L344: please provide the composition of HTF: did it contain glucose? why were sperm subjected to capacitation prior to subjecting to freeze drying?

L404: what does ‘the sperm suspension was replaced every 30 min during ICSI’ mean: does this refer to preparing a freshly rehydrated sample. i.e., FD sperm was used within 30 min after rehydration in all cases?

**Do you want your identity to be public for this peer review?** For information about this choice, including consent withdrawal, please see our Privacy Policy

Reviewer #1: No

---

## [Author Response · Author response to Decision Letter 1]

9 Sep 2025

Response to Editor and Reviewer

Editor

Major comment:

Overall, the discussion on the potential mechanism of the surprising finding is somewhat shallow. Would cellular organelle structures be affected by faster rehydration? Are there studies of rehydrating other materials, microorganisms or even plants, that can provide some clues on the mechanisms?

Response: Thank you for your comment. We acknowledge the limitations in our initial discussion of the underlying mechanisms. We agree that cellular structures, including organelles and membranes, may be influenced by the rate of rehydration. Although we did not investigate this aspect in the present study, we have now expanded the Discussion section to cite studies showing that rehydration conditions affect damage and structural preservation in liposomes, microorganisms, and seeds. This addition provides a broader perspective on how rapid rehydration may alleviate subcellular damage.

Lines: 225-234

Previous research on various organisms and organelles, such as microorganisms, seeds, and liposomes, has shown that rehydration conditions influence recovery from a dried state (5 new references). For example, rapid rehydration may cause osmotic shock and/or sudden membrane phase transitions, both of which can induce rehydration-specific damage, including membrane rupture, organelle degeneration, and oxidative stress. We therefore predicted that our rehydration protocol could cause severe FD sperm damage. However, because FD sperm are directly injected into oocytes by ICSI, membrane or mitochondrial damage may not affect fertilization or embryo development. Furthermore, rapid rehydration may influence the DNA or protamine content of FD sperm nuclei.

We added following 5 new references

33. Arav A, Natan D. Freeze drying of red blood cells: the use of directional freezing and a new radio frequency lyophilization device. Biopreserv Biobank. 2012;10(4):386-94. doi: 10.1089/bio.2012.0021. PubMed PMID: 24849889.

34. Arellano-Ayala K, Lim J, Yeo S, Bucheli JEV, Todorov SD, Ji Y, et al. Rehydration before Application Improves Functional Properties of Lyophilized Lactiplantibacillus plantarum HAC03. Microorganisms. 2021;9(5). Epub 20210508. doi: 10.3390/microorganisms9051013. PubMed PMID: 34066743; PubMed Central PMCID: PMCPMC8150888.

35. Crow LM, Womersley C, Crowe JH, Reid D, Apple L, Rudolph A. Prevention of fusion and leakage in freeze-dried liposomes by carbohydrates. Biochimica et Biophysica Acta. 1986;861:131-40.

36. Makeen MA, Normah MN, Dussert S, Clyde MM. The influence of desiccation and rehydration on the survival of polyembryonic seed of Citrus suhuiensis cv. limau madu. Scientia Horticulturae. 2007;112:376-81.

37. Zhang W, van Winden EC, Bouwstra JA, Crommelin DJ. Enhanced permeability of freeze-dried liposomal bilayers upon rehydration. Cryobiology. 1997;35(3):277-89. doi: 10.1006/cryo.1997.2050. PubMed PMID: 9367615.

Minor comments:

Brackets of references: some are separated by a space from the preceding words, others are not. Please look at the Journals requirements and be consistent.

Response: Thank you for pointing this out. We have corrected all references in the revised version of the manuscript.

L58: change “F” to “f”.

Response: Thank you for pointing this out. In the revised manuscript, we have corrected the capital “F” to a lowercase “f.”

Line: 57

when the first successful freeze-drying of mouse sperm was demonstrated7,

L69: “FD pretreatment” is not clear. Is FD the pretreatment? If so, the word “pretreatment” is not needed. If not, please give an example of the pretreatment using “such as”.

Response: Thank you for this comment. In the revised manuscript, we have replaced “pretreatment” with “treatment.”

Lines: 68-69

Furthermore, sperm subjected to FD treatment showed improved resistance to extreme temperatures ranging from −196°C to 150°C.

L72: “over 200 years”, is this a mistake? I don’t believe that the International Space Station is that old.

Response: Thank you for your comment. “Over 200 years” refers to a theoretical estimate. During the ISS experiment, FD sperm was stored for 6 years. Based on comparisons with X-ray irradiation experiments on Earth, the authors determined that FD sperm could theoretically be preserved for 200 years on the ISS (Wakayama et al., Sci Adv. 2021). To avoid confusion, the text has been revised as follows:

Lines: 69–72

FD sperm stored on the International Space Station showed no reduction in fertilization capacity or developmental potential after 6 years of storage, despite exposure to high levels of radiation

L76: “one-third” of “fresh or cryopreserved sperm”. This implies that fresh and cryopreserved sperm have the same birth rates. Is this correct? If not, please provide two fractions.

Response: Thank you for confirming this point. As reported in previous studies (see references no. 13, 21, and 22), the birth rates of mouse embryos derived from fresh and cryopreserved sperm are comparable. We therefore believe that the current expression is appropriate. In the revised manuscript we have added an additional reference (Ushigome et al., Scientific Reports 2025, 15:30994.) to further demonstrate the validity of this statement.

30. Ushigome N, Wakayama S, Yamaji K, Ito D, Ooga M, Wakayama T. Production of offspring from vacuum-dried mouse spermatozoa and assessing the effect of drying conditions on sperm DNA and embryo development. J Reprod Dev. 2022;68(4):262-70. Epub 20220607. doi: 10.1262/jrd.2022-048. PubMed PMID: 35676029; PubMed Central PMCID: PMCPMC9334318.

L80: please insert the reference of your prior publication.

Response: Thank you for your suggestion. We have added references to our prior publications (i.e., Kamada et al., 2025; Kamada et al., 2018; and Wakayama et al., 1998) to the revised text.

L90: “first sperm freezing”? It seems “freezing” alone is more appropriate.

Response: Thank you for this comment. To avoid possible misunderstanding, we have revised “first sperm freezing” to “freezing.”

Line: 88

During the first sperm freezing,

L89-98: this section confuses the readers. First FD is described as a 4-step procedure, but then it was stated that drying without freezing can also be done. These are conflicting statements. Please revise.

Response: Thank you for your comment. We apologize for the unclear description. We have revised it as follows.

Lines: 93–95

To investigate the effects of the drying process, we conducted an experiment in which sperm were directly vacuum-dried without freezing. This experiment demonstrated that…

L126: please give some details of the filter paper because not all filter papers are the same.

Response: Thank you for your comment. We have added the catalog number to the revised version of the text.

Line: 118

(Merck Millipore Low Extractable Filters GSWP09000)

L256: “structural changes”, it is unclear what structure is being discussed here, plasma membrane, organelles or DNA? The following statements also need to be clearer in the specific structures being discussed.

Response: Thank you for this suggestion. We have revised the section to explicitly reference “DNA structural changes.”

Lines: 250–255

As FD treatment alters the sperm plasma membrane and can cause DNA damage, structural changes to DNA are likely. Once dried sperm are rehydrated, inconsistencies may occur between surface DNA regions, which recover quickly, and central regions, which remain dry during early water infiltration. Such inconsistencies may lead to chromosomal damage and exacerbate minor DNA lesions caused by FD processing.

Reviewer 1

General comments:

This paper describes effects of different rehydration rates on DNA damage and fertilization potential of freeze-dried (FD) mouse sperm. More specifically, FD sperm were rehydrated using HTF medium instead of water or water supplemented with PVP for decreasing the infiltration speed, while water of increasing temperatures was used for obtaining more rapid rehydration. The comet was employed to assess sperm DNA damage after rehydration. In addition, after fertilization, gamma-H2A.x immunostaining and abnormal chromosome segregation patterns were investigated. ICSI with FD sperm was performed to determine fertilization potential and in vitro development of sperm rehydrated using different protocols, while embryo transfers with obtained 2-cell stages were used to assess if these could lead to producing offspring. It was found that rehydration of FD-sperm with water containing increasing PVP concentrations was damaging. Furthermore, rapid rehydration of FD sperm, i.e., with water of 50C, resulted in decreased sperm DNA damage, which in turn increased ICSI outcomes and birth rates. Rehydration with water of 70 or 90C resulted in lower sperm DNA damage, but resulted in the need of artificial oocyte activation following ICSI.

Response: Thank you for reading our manuscript in detail and for your constructive comments. We have incorporated your recommendations into the revised version of the manuscript.

Major comments:

1) This is a full paper, in which multiple aspects were investigated; focusing on effects of rehydration-related damage of FD sperm.

Although it may not be unexpected that sperm biomolecular damage progresses upon extended rehydration duration, due to accumulation of reactive oxygen species, and use of high PVP concentrations with ICSI impairs development; it is important to document these data.

These authors published several papers on freeze-drying of mouse sperm, and here merely use their established protocols and assays while testing a further variable.

Anyways, the authors present a lot of work, with testing multiple concentrations/treatments resulting in dose-dependent effects due to both slower and faster rehydration, while correlating sperm DNA damage with ICSI outcomes as well as offspring when used for embryo transfer.

Response: Thank you for your valuable comment. We are pleased that you have gained a thorough understanding of the research presented in this paper.

2) The rationale of several methodologies is not clear.

Why was HTF used as a base medium for freeze drying, without use of EDTA, antioxidants and/or disaccharides as has been described beneficial by others?

Response: Thank you for this comment. In this study, we particularly examined the effect of rehydration alone; therefore, FD sperm were produced using a simple HTF medium developed in our laboratory as a base medium. To make this clearer, the following text has been added to the revised version of the main text.

Lines 336–338

In this study, to examine the effect of rehydration on FD sperm, we decided to produce FD sperm in an HTF medium that did not contain freeze-drying protectants.

Was the duration between starting rehydration and performing ICSI kept constant, i.e., was ICSI performed after varying durations in case of slow vs rapid rehydration and can differences simply be explained by the time sperm spend in their rehydration medium?

Response: Thank you for your question. In this study, an ampoule from each experimental group was opened, rehydrated, and ICSI was performed immediately. Therefore, the elapsed time between the start of rehydration to ICSI was held constant. To make this clearer in the revised manuscript we have added the following text:

Lines 390–392

Ampoules from each experimental group were opened, rehydrated, and ICSI was performed immediately. Therefore, the elapsed time between the start of rehydration and ICSI was constant for all experiments.

Were sperm washed after rehydration, i.e., for avoiding interference of different types and concentrations of compounds with the post-rehydration assays performed (comet assay, ICSI)?

Response: Thank you for this comment. We did not perform washing after rehydration. Residual compounds were present in minimal amounts that were unlikely to affect the assays. Additionally, we wished to avoid potential damage that washing procedures might cause. To clarify this, the following text has been added to the main text:

Line: 394

For each sperm microinjection, 1–2 μL of sperm suspension was transferred directly to the injection chamber without washing.

Line: 418

Specimens were then mounted on slides without washing,

3) It is appreciated that the authors aimed to validate that their different approaches result in different rehydration rates, however, infiltration in a paper strip does not reflect water movement into cellular structures (i.e., containing membranes/organelles and compartments with different compositions). The authors note this. Anyways, describing physical properties of the rehydration solutions used (osmolality, viscosity, temperature) may be sufficient here, and the description in L119-136 can be shortened/presented more condensed (and supplemental figure 1 can be omitted).

Response: Thank you for this comment. We agree that infiltration into filter paper does not directly reflect water movement into cells. However, we conducted this experiment to highlight potential differences in the physical properties (e.g., osmotic pressure, viscosity, temperature) of the rehydration solutions. Following your suggestion, we have shortened this section and removed Supplementary Fig. 1.

Lines: 123–128

We added droplets of HTF medium, PVP solutions at varying concentrations, and ultrapure water at different temperatures onto filter paper and recorded the diffusion distance per unit time. HTF medium spread more slowly than ultrapure water, PVP solutions showed concentration-dependent diffusion that was consistently slower than water, and ultrapure water spread faster as temperature increased (S1 Table).

4) Providing tables with numbers on actual numbers of sperm/oocytes/embryos studies as supplemental data is good. Supplemental figure 2, however, is maybe better presented in the main document. Why did the authors decide presenting these data in the supplement or can this be moved?

Response: Thank you for this suggestion. We have moved Supplemental Figure 2 into the main text as the new Figure 3.

5) The discussion section is relatively long, and possibly can be shortened while focusing on the major findings and omitting repetitions. Also the introduction can possibly presented in a more condensed manner. The reference listing contains relatively many papers from the same authors.

Response: Thank you for this comment. In the revised manuscript, we have shortened the Discussion and parts of the Introduction by removing redundant text. Regarding citations, it should be noted that our laboratory was the first to produce offspring from FD sperm (Wakayama and Yanagimachi, Nature Biotechnology 1998). Since this has remained the major focus of our research, citing many of our past studies is necessary for accurate comparison. We hope you understand this context.

Delated from line 49 to 52, from 270 to 281, from 289 to 296, from 290 to 295.

49~52:

handling liquid nitrogen is challenging and poses risks of frostbite, asphyxiation, and rapid gas expansion. Moreover, it is also necessary to frequently replenish liquid nitrogen to keep temperatures low

270~281:

However, while a comet assay showed the lowest level of DNA damage in FD sperm samples rehydrated in water at 70℃, the incidence of ACS in two-cell stage embryos was lowest when the rehydration water was 50℃, and this value increased at temperatures above 70℃ compared to the RT control. We note that the comet assay is most suitable for visualizing minor DNA damage and for evaluating the total amount of DNA damage, whereas the ACS assay is better suited for detecting the presence of severe DNA damage. Taken together, these results indicate that increasing the temperature of the ultrapure water used for rehydration can accelerate infiltration speed into the nucleus, thereby reducing minor DNA damage as the temperature increases. However, although we observed decreases in the level of severe DNA damage at temperatures between 37℃ and 50℃, we also note that the level of damage can increase when temperatures exceed this range.

289-296:

On the other hand, FD sperm could produce offspring even after brief exposures to temperatures as high

---

## [Decision Letter · Decision Letter 1]

17 Sep 2025

Improved Birth Rates via Rehydration of Mouse Freeze-Dried Spermatozoa using High-Temperature Ultrapure Water

PONE-D-25-31910R1

Dear Dr. Wakayama,

We’re pleased to inform you that your manuscript has been judged scientifically suitable for publication and will be formally accepted for publication once it meets all outstanding technical requirements.

Kind regards,

Xiuchun Tian

Academic Editor

PLOS ONE

Additional Editor Comments (optional):

Reviewer #1:

Reviewers' comments:

Reviewer's Responses to Questions

**Comments to the Author**

Reviewer #1: All comments have been addressed

2. Is the manuscript technically sound, and do the data support the conclusions?

Reviewer #1: Yes

3. Has the statistical analysis been performed appropriately and rigorously?

Reviewer #1: I Don't Know

4. Have the authors made all data underlying the findings in their manuscript fully available?

Reviewer #1: Yes

5. Is the manuscript presented in an intelligible fashion and written in standard English?

Reviewer #1: Yes

Reviewer #1: This reviewer has reviewed the original manuscript in detail.

Following feedback given to their original submission, the authors have submitted a revised manuscript and provided an accompanying letter in which they replied to the issues raised by the reviewers in detail. Also, they listed all changes they made in their revised submission.

This reviewer has now read the revised manuscript, and believes that the authors improved their paper via incorporating the feedback given to them. Furthermore, in their rebuttal letter, the authors have addressed all issues and questions raised by this reviewer. I have no further comments.

**Do you want your identity to be public for this peer review?** For information about this choice, including consent withdrawal, please see our Privacy Policy

Reviewer #1: No

---

## [Editor Report · Acceptance letter]

PONE-D-25-31910R1

PLOS One

Dear Dr. Wakayama,

I'm pleased to inform you that your manuscript has been deemed suitable for publication in PLOS One. Congratulations! Your manuscript is now being handed over to our production team.

Kind regards,

on behalf of

Dr. Xiuchun Tian

Academic Editor

PLOS One